# BRIDGE: A Risk-Aware Framework for Evaluating Behavioral Fidelity in LLM Agents

## Abstract

We study behavioral fidelity: whether a conversational agent reliably enacts intended aims. While recent work evaluates accuracy or task success, it offers limited tools for calibrated behavioral assessment in multi-turn dialog. We present BRIDGE, a selective evaluation framework with deterministic gates for validity and safety, a probabilistic judge for trait scoring, and a conformal-inspired mapper that converts scores to {yes, ambiguous, no} using per-trait dual thresholds centered at empirical human prevalence with half-width $(1 - \kappa)/2$, where $\kappa$ is interrater agreement. A dialog passes only if all relevant traits are yes; otherwise, it abstains and escalates. Thresholds are derived from a small, disjoint, human-annotated calibration set. On 5,960 dialogs across two domains and two aims, 87.7% pass without human review, and abstentions concentrate in semantically appropriate traits. Stored thresholds match the declared rule to machine precision. Additional experiments on 61 organic conversational recommender dialogs (CRS-61) show that gates remain selective on real logs and that pass/abstain decisions are largely stable under Judge prompt paraphrases.

## 1 Introduction

Large language models are deployed as conversational agents across travel, lifestyle, support, finance, and enterprise settings. Beyond factual accuracy, users care about *behavioral fidelity*: whether an agent reliably enacts intended aims such as being transparent and cognitively supportive (EDUCATIVE) or enabling curiosity, discovery, and serendipity (EXPLORATIVE). Modern networks can be miscalibrated (Guo et al., 2017; Minderer et al., 2021; Ovadia et al., 2019), and selective evaluation motivates explicit abstention under uncertainty (Chow, 1970; Geifman & El-Yaniv, 2017; 2019), suggesting a need for auditable, behavior-centric evaluation.

This paper presents **BRIDGE**[1], a domain-agnostic framework for selective evaluation of behavioral fidelity in conversational agents. Deterministic *gates* enforce validity and safety, a probabilistic *Judge* scores task-specific traits, and a *conformal-inspired mapper* converts trait scores to {yes, ambiguous, no} using per-trait dual thresholds *centered at empirical human prevalence* with *half-width* $(1 - \kappa)/2$ after clipping, where $\kappa$ is interrater agreement. The *system passes* a dialog only if *all* relevant traits are yes; otherwise it *abstains* and escalates. Thresholds are set once on a separate human slice; evaluation is disjoint; and we release numeric artifacts (thresholds, tables, and figure or metric scripts) for audit. We make *process guarantees*, not coverage guarantees; the interface is compatible with conformal risk control when formal pass-risk control is desired (Angelopoulos et al., 2024; Romano et al., 2020; Overman et al., 2024).

**Contributions.**

- **Selective behavioral evaluation.** We cast aim adherence as selective evaluation with deterministic gates for validity and safety, a probabilistic *Judge* for trait scoring, and a conformal-inspired mapper that yields yes/ambiguous/no via per-trait dual thresholds centered at empirical human prevalence with half-width $(1 - \kappa)/2$.
- **Human-anchored calibration with process guarantees.** A disjoint human calibration slice provides prevalence and agreement estimates used to set and freeze thresholds, with evaluation performed on a separate set of dialogs. We release auditable numeric artifacts (thresholds JSON,

---

[1] Behavioral Role Integrity via Dual lane Guardrails and Evaluation.

summary tables, and figure or metric scripts), yielding process guarantees about disjoint calibration, frozen thresholds, and deterministic mapping. The interface remains compatible with conformal risk control methods that can, in future deployments, be wrapped around BRIDGE to bound pass risk.

- **Domain-agnostic instantiation.** We demonstrate BRIDGE on EDUCATIVE and EXPLORATIVE aims in two synthetic domains (Travel, Lifestyle) and on a small organic conversational recommender slice (CRS-61), and we outline how to port to other agentic tasks by swapping the trait schema while reusing the calibration protocol.

## 2 BACKGROUND

**Behavioral evaluation beyond accuracy.** Modern neural networks can be confidently wrong and poorly calibrated, which undermines behavior-level assessment when evaluation relies on raw confidence (Guo et al., 2017; Minderer et al., 2021; Ovadia et al., 2019; Lakshminarayanan et al., 2017). This motivates selective evaluation that exposes uncertainty and routes hard cases for review, rather than optimizing a single accuracy metric, and aligns with calls to evaluate procedures and behaviors in addition to end metrics (Lovering et al., 2022).

**Selective prediction and abstention.** Reject-option classification trades coverage for reliability (Geifman & El-Yaniv, 2017). Deep selective methods formalize the risk–coverage curve and introduce architectures that learn when to abstain (Geifman & El-Yaniv, 2019). BRIDGE follows this paradigm but ties abstention to *human agreement*: per-trait bands are centered at empirical prevalence with half-width $(1 - \kappa)/2$ after clipping, producing auditable thresholds and safety margins. Closely related work proposes selective judging with provable human-agreement targets and cascaded escalation (Jung et al., 2025).

**Conformal methods and risk control.** Conformal prediction gives distribution-free, finite-sample guarantees for set outputs; adaptive and regularized variants enable practical classification sets such as APS and RAPS (Romano et al., 2020). Conformal Risk Control targets expected monotone risks under coverage constraints with abstention and offers a wrapper when formal pass-risk control is required (Angelopoulos et al., 2024). Our mapper is *conformal inspired*: thresholds are fixed from a separate human slice and applied post hoc; we make *process guarantees* in this study and note compatibility with CRC without changing the operator workflow.

**LLM judges: reliability and calibration.** LLM-as-a-judge can correlate with human preferences, but evaluators may recognize and favor their own generations and exhibit sensitivity to survey-style artifacts (Panickssery et al., 2024; Dominguez-Olmedo et al., 2024). We therefore treat the judge as an *auditable instrument*: a small human slice yields empirical prevalence and interrater agreement to set per-trait bands; the system passes only if all relevant traits are yes, otherwise it abstains. Recent evidence on personalized preference following further motivates anchoring to human agreement rather than raw judge confidence (Zhao et al., 2025).

**Learning to defer, human expertise, and routing.** Learning to defer jointly trains a predictor and a deferral policy to route examples to experts when the model is uncertain or biased (Madras et al., 2018; Mozannar & Sontag, 2020). These methods can reduce system-level risk when expert traces exist. BRIDGE is complementary: a rule-based, human-anchored deferral that uses only trait annotations, with the same interface admitting a learned deferrer later if desired. Modeling preference diversity further justifies using agreement to set conservative bands (Bakker et al., 2022). Human expertise often adds value precisely where algorithms are least informative, reinforcing the operational stance to escalate rather than overclaim in high-stakes settings; the abstention queue surfaces such cases explicitly for targeted review (Alur et al., 2024).

**Multi-turn agents and interactive benchmarks.** Recent benchmarks stress multi-step agents in realistic environments: *MINT* studies multi-turn tool-using interactions (Wang et al., 2024); *AgentBoard* unifies multi-round, partially observable tasks with fine-grained metrics (Chang et al., 2024); *WebArena* evaluates browser agents on open-web goals (Zhou et al., 2023); *WebShop* targets grounded web interaction at scale (Yao et al., 2022); and *SWE-bench* measures code-agent performance on real GitHub issues (Jimenez et al., 2023). These works emphasize end-to-end task success

and tool competence; our focus is complementary: role-level *behavioral fidelity* with explicit abstention and human-anchored calibration. BRIDGE uses a conversational recommendation testbed as one example but is not tied to any single domain.

**Summary.** Compared to judge-only or task-completion benchmarks, BRIDGE operationalizes *behavioral fidelity* via explicit aims and traits, and introduces a calibrated, abstaining *gate–judge–mapper* evaluation that surfaces uncertainty and routes hard cases for review while remaining compatible with conformal risk control.

## 3 METHODS

**Objective.** We evaluate whether a conversational agent reliably *enacts intended behaviors* in multi-turn interaction. We target two structured aims (EDUCATIVE, EXPLORATIVE) and measure *behavioral fidelity*: the degree to which a dialog satisfies the trait set associated with the instructed aim.

### 3.1 PROBLEM SETUP

Let $x_{1:T}$ denote user turns and $y_{1:T}$ agent turns in strict alternation with fixed depth $T=12$. Let $a \in \{\text{EDUCATIVE}, \text{EXPLORATIVE}\}$ be the instructed aim, and $K_{1:T}$ the retrieved knowledge available to the agent at each agent turn. The agent policy emits $y_t^{(r)}$ at temperature $\tau \in \{0.0, 0.3, 0.5, 0.7, 1.0\}$, with $r \in \{1, 2, 3\}$ *independent replicates per (seed, aim, temperature)*. We define trait sets

$$\mathcal{T}_{\text{EDUCATIVE}} = \{\textit{transparency/explanation}, \textit{cognitive support}, \textit{self-reflection}\},$$
$$\mathcal{T}_{\text{EXPLORATIVE}} = \{\textit{curiosity}, \textit{discovery/novelty}, \textit{serendipity}\},$$

which are observable in agent behavior and used for gate filtering, Judge scoring, and decision making.

**Calibration and decision rule (core).** From a disjoint human slice, for each trait $t$ we compute empirical prevalence $\text{prev}_t$ and inter-annotator agreement (Cohen's $\kappa_t$), and define $\alpha_t = 1 - \kappa_t$. Dual thresholds are

$$\text{lower}_t = \text{clip}\left(\text{prev}_t - \tfrac{\alpha_t}{2}, 0, 1\right), \qquad \text{upper}_t = \text{clip}\left(\text{prev}_t + \tfrac{\alpha_t}{2}, 0, 1\right). \tag{1}$$

**Conventions.** $\text{clip}(z, 0, 1) = \max\{0, \min\{1, z\}\}$.

Given Judge score $\hat{p}_t \in [0, 1]$, the mapper assigns

$$\mathcal{C}_t = \begin{cases} \text{YES}, & \hat{p}_t \geq \text{upper}_t, \\ \text{NO}, & \hat{p}_t \leq \text{lower}_t, \\ \text{AMBIGUOUS}, & \text{otherwise.} \end{cases} \tag{2}$$

**Decision.** $D_a = \mathbb{1}\left[\bigwedge_{t \in \mathcal{T}_a} (\mathcal{C}_t = \text{YES})\right], \qquad A_a = 1 - D_a$ (pass $D_a{=}1$; abstain/escalate $A_a{=}1$).

We also report an aim-level fidelity score $F_a = \min_{t \in \mathcal{T}_a} \hat{p}_t$. The nearest-bound safety margin is $m_t = \min(\text{prev}_t - \text{lower}_t, \text{upper}_t - \text{prev}_t).^2$

### 3.2 GATE–JUDGE–MAPPER PIPELINE

The BRIDGE pipeline (Fig. 1, Alg. 1) combines four deterministic gates (G1–G4), a probabilistic Judge, and a conformal-inspired mapper. It makes uncertainty explicit and *escalates* when any trait is not YES. We offer *process guarantees*: thresholds are frozen from a disjoint human slice and applied post hoc. Although we do not claim conformal prediction (CP) or coverage–risk control (CRC) guarantees, the interface supports CP/CRC-style wrappers when needed.

---

[2]When clipping does not bind, band width equals $1 - \kappa_t$ and half-width equals $(1 - \kappa_t)/2$.

---

**Algorithm 1** Selective evaluation pipeline (BRIDGE)

---

**Require:** $(x_{1:T}, y_{1:T}, K_{1:T})$, aim $a$, trait set $\mathcal{T}_a$, frozen thresholds $\{(\text{lower}_t, \text{upper}_t)\}_{t \in \mathcal{T}_a}$ with version hash $h$

**Ensure:** One terminal outcome in {PASS, ABSTAIN_GATE, ABSTAIN_MAPPER} recorded with audit metadata

1: **Config check.** Assert $\mathcal{T}_a \neq \varnothing$; load thresholds read-only (hash $h$).
2: **Gates (G1–G4).**
3: **for** $g \in \{\text{G1}, \text{G2}, \text{G3}, \text{G4}\}$ **do**
4:    **if** $g$ fails **then**
5:       **emit** ABSTAIN_GATE with reason code; **record**; **return**
6:    **end if**
7: **end for**
8: **Judge.** Compute $\hat{p}_t \in [0, 1]$ for all $t \in \mathcal{T}_a$; if any score is missing, emit ABSTAIN_MAPPER (T_MISS); record; return.
9: **Mapper.** For each $t$, map $\hat{p}_t \mapsto \mathcal{C}_t \in \{\text{YES, AMBIGUOUS, NO}\}$ via Eq. equation 2.
10: **if** any $\mathcal{C}_t \in \{\text{AMBIGUOUS, NO}\}$ **then**
11:    emit ABSTAIN_MAPPER with trait summary; record; return
12: **else**
13:    Compute $F_a = \min_{t \in \mathcal{T}_a} \hat{p}_t$; emit PASS with labels and $F_a$; record; return
14: **end if**

---

**Abstention logic and escalation protocol.** On any gate failure, the system immediately abstains and logs an escalation with a canonical reason code (G1_RETRIEVAL_INSUFF, G2_CONTRADICTION, G3_ACT_MISS, G4_TONE_BREACH) and relevant evidence. The Judge and mapper are skipped. If all gates pass but any trait maps to AMBIGUOUS or NO, the system abstains with T_AMBIG:<t> or T_NO:<t> as justification.

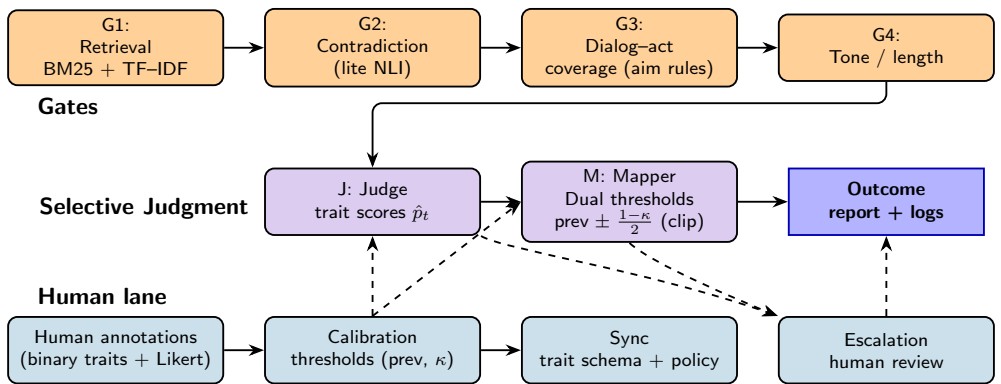

Figure 1: BRIDGE pipeline. Top: deterministic gates (G1–G4) check evidence sufficiency, contradiction, act presence, and tone/length. Middle: a probabilistic Judge and conformal-inspired mapper assign trait-level labels. Dashed arrows indicate calibration and escalation. A dialog passes only if all relevant traits are yes; otherwise the system abstains.

**G1–G4 gate logic.** **G1 retrieval sufficiency (epistemic):** agent claims in $y_{1:T}$ must be supported by $K_{1:T}$ via standard IR coverage (e.g., BM25) (Robertson et al., 2009; Manning, 2008). **G2 contradiction screen (logical):** lightweight NLI checks for claim–evidence mismatch; contradictions trigger abstention (Bowman et al., 2015; Nie et al., 2019). **G3 dialog-act coverage (intent):** rule-based detectors verify presence of aim-required acts (e.g., *explain*, *step-by-step support* for EDUCATIVE; *novelty cue*, *branching probe* for EXPLORATIVE). The Judge evaluates quality (Stolcke et al., 2000). **G4 tone/verbosity guard (pragmatic):** style and length filters ensure supportive tone and limit over-verbosity. Dialogs passing all gates are scored by the Judge to produce $\hat{p}_t$, then mapped to YES/AMBIGUOUS/NO. Gates certify formal validity; the Judge and mapper determine trait quality.

### 3.3 EVALUATION PROTOCOL

We generate 6,000 twelve-turn dialogs across two domains (Travel, Lifestyle). For each domain we curate 100 user seeds and evaluate both aims across five temperatures $\{0.0, 0.3, 0.5, 0.7, 1.0\}$

with three independent replicates per condition and retrieval at every agent turn (standard in RAG-style systems (Lewis et al., 2020)), yielding $100 \times 5 \times 3 \times 2 = 3{,}000$ dialogs per domain. In addition, for an organic case study we evaluate BRIDGE on 61 anonymized sessions from a deployed conversational recommender system (CRS-61), applying the same Gate–Judge–Mapper pipeline and frozen thresholds without re-tuning.

**Calibration and inter-rater reliability.** A separate human slice ($n{=}40$ dialogs; two raters) provides, for each trait, empirical prevalence and inter-annotator agreement $\kappa$. We report Cohen's $\kappa$ and ICC$(2,1)$ with 95% confidence intervals (Shrout & Fleiss, 1979). Per-trait dual thresholds follow Eq. equation 1 after clipping. Lower $\kappa$ widens bands and increases abstention rather than over-claiming. **Leakage prevention.** Calibration uses a disjoint human slice $\mathcal{H}$; its dialogs, seeds, and rater labels do not appear in evaluation. Thresholds derived from $\mathcal{H}$ are serialized (JSON with hash) and *loaded read-only* during evaluation. Generation prompts, retrieval index, and Judge parameters are fixed *a priori* and versioned; no tuning is performed on evaluation dialogs.

**Statistical analysis.** We report pass and abstention rates with exact Clopper–Pearson 95% confidence intervals (Clopper & Pearson, 1934), aggregating counts at the *seed* level to reduce intra-seed dependence.[3]

**Terminology and baselines.** A *turn* is one message; each dialog has twelve turns (six user, six agent). The *agent* is the LLM-generated agent; the *system* is the full BRIDGE pipeline. We compare to: (i) *Judge only* (no gates or mapper), (ii) *No mapper* (hard single cutoff without calibration), and (iii) *No gates* (Judge and mapper only).

### 3.4 COMPUTATIONAL FOOTPRINT

To support reproducibility and transparency, we report the compute environment, API usage, and associated costs for evaluation. All runs used a MacBook M1 (16 GB RAM; macOS 15.7). Only the **Judge** called an external API (*OpenAI* `gpt-4.1-2025-04-14`, deterministic decoding: temperature$= 0$, top_p$= 1$, $n{=}1$); retrieval, gates, and the mapper ran locally on CPU.

**Billing snapshot (2025-09-18, 00:00–24:00 UTC).** Requests: *25,874*[4]. Input tokens: *11,730,104*. Output tokens: *77,622*. Total cost: *$21.706312*. Outputs were *0.66%* of total tokens, so costs are input-dominated. *Scope note:* this snapshot covers *all* Judge traffic on that UTC day, including calibration passes, smoke/health checks, automatic re-tries on transient errors (HTTP 429/5xx with exponential backoff), and a small number of abstention-triggered re-checks. It therefore exceeds the final-count dialogs evaluated that day. **Derived quantities (exact).** Average tokens/request: $\frac{11{,}730{,}104+77{,}622}{25{,}874} = 456.35$. Cost/request: $\frac{21.706312}{25{,}874} = 8.389{\times}10^{-4}\,\$ \approx \$0.00084$. Effective \$ per 1K input tokens: $\frac{21.706312}{11{,}730.104} = \$0.00185$. (Per 1K total tokens: \$0.00184.) **Per-dialog Judge policy and cost.** Our default evaluation uses *one* Judge call per dialog; an additional call is issued only on explicit abstention (rare) or transient API failure (auto-retry with jitter; capped at one retry). Under this policy the expected Judge cost is $\approx \$0.00084$ per dialog (\$5.03 for 6,000 dialogs). Two and three Judge calls per dialog would scale to \$10.07 and \$15.10 for 6,000 dialogs, respectively. Local components were CPU-only and vectorized; wall time was dominated by Judge API latency. All figures refer specifically to `gpt-4.1-2025-04-14` and are computed directly from provider billing.

## 4 RESULTS

**Evaluation setup.** We study four cells (Travel–Educative, Travel–Explorative, Lifestyle–Educative, Lifestyle–Explorative), each with 1,500 target dialogs ($N{=}6{,}000$). As specified in §3, dialogs are filtered by G1–G4 and then evaluated by the Judge and conformal-

---

[3]Temperature sweeps affect output variability and exploration in LLM decoding (Holtzman et al., 2019).

[4]Of the 25,874 calls, only those linked to dialog IDs contribute to reported metrics; calibration/smoke/retry calls are excluded from evaluation counts.

inspired mapper (Eqs. 1–2); a dialog *passes* only if all relevant traits are `yes`, else it *abstains* (escalation). For summary reporting we use the session-level fidelity (minimum trait score).[5]

## 4.1 GATE OUTCOMES (G1–G4)

G1 removed 17 Travel–Educative dialogs; G3 removed 23 Lifestyle–Explorative dialogs; G2 found 0 contradictions; G4 found 0 violations. This yields $N=5{,}960$ dialogs admitted to the Judge, as shown in Table 1.

Table 1: Flow of dialogs through evaluation gates (G1–G4), by condition. G1 checks retrieval sufficiency; G3 enforces dialog-act coverage. G2 (contradiction screen) and G4 (tone/verbosity guard) removed no dialogs and are omitted for brevity. Only dialogs admitted by all gates proceed to the Judge.

| Condition | Start | G1 removed | G3 removed | Admitted to Judge |
|---|---|---|---|---|
| Travel–Educative | 1500 | 17 | 0 | 1483 |
| Travel–Explorative | 1500 | 0 | 0 | 1500 |
| Lifestyle–Educative | 1500 | 0 | 0 | 1500 |
| Lifestyle–Explorative | 1500 | 0 | 23 | 1477 |

## 4.2 SELECTIVE JUDGE WITH CONFORMAL-INSPIRED ABSTENTION

Table 2 shows outcomes of the probabilistic–thresholding pipeline for $N=5{,}960$. Decisions follow the selective rule: `abstain` indicates escalation to humans. *Counts are dialog-level; pass-rate confidence intervals are computed as exact Clopper–Pearson at the* seed *level: per (domain, aim), each seed contributes the proportion of its* 15 *dialogs (5 temperatures × 3 replicates) that pass, and CIs are then reported over the* 100 *seeds.*

Table 2: Judge decisions by condition. We report $N$, `Pass`/`Abstain` counts, and exact Clopper–Pearson 95% CIs for the seed-level pass rate (100 seeds per condition; 15 dialogs per seed).

| Condition | $N$ | Pass | Abstain | Pass % [95% CI] |
|---|---|---|---|---|
| Travel–Educative | 1483 | 948 | 535 | 63.92 [61.45, 66.33] |
| Travel–Explorative | 1500 | 1495 | 5 | 99.67 [99.22, 99.86] |
| Lifestyle–Educative | 1500 | 1392 | 108 | 92.80 [91.38, 94.00] |
| Lifestyle–Explorative | 1477 | 1390 | 87 | 94.11 [92.79, 95.20] |

**Pass fidelity (evidence strength).** Among `passed` dialogs, the session-level fidelity (minimum of the three relevant trait scores) is *higher* in explorative settings and *moderate* in educative ones: *Travel–Explorative*: median 0.7 [p25 0.5, p75 0.7] ($n=1495$); *Lifestyle–Explorative*: median 0.7 [0.5, 0.7] ($n=1390$); *Travel–Educative*: median 0.5 [0.5, 0.5] ($n=948$); *Lifestyle–Educative*: median 0.5 [0.5, 0.5] ($n=1392$). Distributions (from frozen CSV exports) indicate that passes clear non-trivial thresholds rather than skirting the margin.

**Where abstentions come from (trait-level analysis).** Abstentions are *targeted*, not random: in *Travel–Educative*, **Self-reflection** contributes to 79.8% of abstaining dialogs and **Cognitive support** to 51.8%; in *Lifestyle–Educative*, **Self-reflection** contributes to 83.3%. For *Lifestyle–Explorative*, **Discovery/novelty** and **Serendipity** contribute to 70.1% and 66.7% of abstentions, respectively; *Travel–Explorative* has only 5 abstentions, predominantly **Serendipity** (80%).[6] This aligns with task semantics: reflective/explanatory depth drives abstentions for educative aims; novelty/serendipity evidence drives abstentions for explorative aims. **Human workload and selective savings.** Selective judging escalates 735/5,960 dialogs (12.3%) to humans; 87.7% require no human review. This reflects the study's fixed operating point (Eq. 1). Exact counts are given in Table 2.

---

[5]Gate failures occur upstream at G1–G4 and are recorded as `abstain_gate`; the Judge/mapper emits only `pass` or `abstain_mapper`. Thresholds are frozen from a disjoint human slice; no post-hoc tuning.

[6]Shares are computed over abstaining dialogs within each condition; multiple traits can trigger escalation in one dialog, so shares need not sum to 100%.

### 4.3 HUMAN RELIABILITY

On an independently rated subset (40 dialogs; 20 educative + 20 explorative in the travel domain), two coders annotated six target traits (binary presence and 1–5 Likert). Human reliability is *moderate to substantial*, with Cohen's $\kappa \approx 0.64$–$0.77$, and absolute agreement is *good* on most traits, ICC(2,1) $\approx 0.56$–$0.87$. *Intervals are computed as specified in §3; full per-trait point estimates and 95% CIs appear in Table 3.* These reliability estimates underpin the calibrated per-trait thresholds used by the mapper.

Table 3: Human reliability by trait ($n=40$ dialogs per trait; two independent raters). We report Cohen's $\kappa$ for binary presence and ICC(2,1) for 5-point Likert ratings, with 95% percentile bootstrap confidence intervals (1,000 resamples). Moderate to substantial agreement supports use of human prevalence and agreement ($\kappa$) to calibrate trait-specific thresholds.

| Trait | $\kappa$ [95% CI] | $\mathbf{ICC(2,1)}$ [95% CI] |
|---|---|---|
| Transparency | 0.697 [0.419, 0.909] | 0.625 [0.403, 0.773] |
| Cognitive_Support | 0.725 [0.358, 1.000] | 0.819 [0.648, 0.919] |
| Self_Reflection | 0.646 [0.398, 0.850] | 0.572 [0.363, 0.710] |
| Discovery | 0.688 [0.254, 1.000] | 0.580 [0.337, 0.744] |
| Curiosity | 0.773 [0.348, 1.000] | 0.828 [0.694, 0.911] |
| Serendipity | 0.667 [0.416, 0.890] | 0.586 [0.343, 0.751] |

### 4.4 CALIBRATION ANALYSIS

Human annotations yield per-trait dual thresholds derived from inter-annotator agreement ($\alpha = 1 - \kappa$) and centered on empirical prevalence (Fig. 2a). Bands narrow where agreement is high and widen where it is low, implementing conservative abstention near decision boundaries. We observe a strong monotone relationship between agreement and band width (Fig. 2b; Pearson $r \approx -0.99$, Spearman $\rho \approx -0.94$), indicating an explicit trade of uncertainty for safety at lower $\kappa$. **Thresholds are frozen from the human annotation set and evaluated on disjoint dialogs; no post-hoc tuning is applied.** Table A.1 (appendix) reports full per-trait statistics: prevalence, Cohen's $\kappa$, $\alpha=1-\kappa$, dual thresholds, and nearest-bound safety margins, ranked in risk-first order. Figures 2a–2b visualize these quantities in the main text.

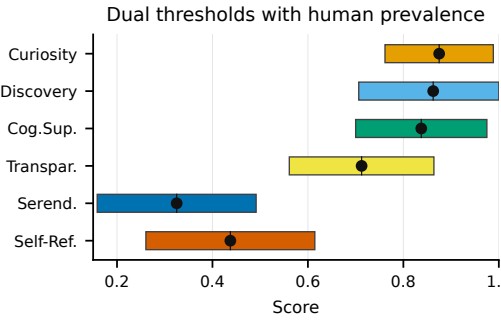
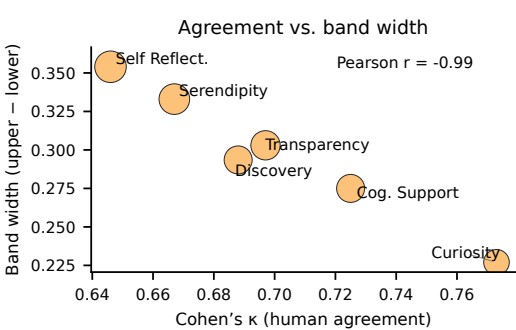

(a) **Dual thresholds with human prevalence.** Bands follow prev $\pm (1 - \kappa)/2$ (clipped); dots mark human prevalence. Narrower bands reflect higher agreement; traits near a bound are treated conservatively.

(b) **Agreement vs. band width (marker size $\propto$ safety margin).** Lower $\kappa$ induces wider bands, trading coverage for safety; Pearson $r \approx -0.99$ *across six traits*.

Figure 2: **Calibration overview.** (a) Per-trait operating envelopes centered on human prevalence; (b) monotone trade-off between human agreement and band width, with marker size encoding nearest-bound safety margin.

### 4.5 ROBUSTNESS AND SENSITIVITY

**Fidelity distributions.** Among `passed` dialogs, explorative aims exhibit higher median fidelity scores with narrower interquartile ranges (median 0.7, IQR [0.5, 0.7]), while educative aims center

at 0.5. This suggests that passed dialogs clear nontrivial thresholds rather than skirting decision boundaries, consistent with the selective evaluation design.

**Trait drivers of escalation.** Abstentions are concentrated in semantically aligned traits. In *Travel–Educative*, **Self-reflection** contributes to 79.8% of abstaining dialogs and **Cognitive support** to 51.8%; in *Lifestyle–Educative*, **Self-reflection** contributes to 83.3%. In *Lifestyle–Explorative*, **Discovery/novelty** and **Serendipity** account for 70.1% and 66.7% of abstentions, respectively; *Travel–Explorative* has only 5 abstentions, predominantly **Serendipity** (80%).[7] These patterns align with the intended semantics of the aims: reflective and explanatory depth for EDUCATIVE, and novelty and serendipity for EXPLORATIVE.

**Threshold design.** We conduct two post-hoc diagnostics using stored trait scores. First, we relax the yes threshold by $\varepsilon \in \{0.00, 0.02, 0.05\}$ while keeping the no cutoff fixed, re-mapping all scores without re-running the model. Across conditions, pass rates vary monotonically and remain stable under small perturbations, demonstrating robustness of abstain/pass decisions.

Second, we simulate a naive baseline that applies a single 0.5 cutoff per trait and passes a dialog only if all three trait scores exceed 0.5, without abstention. This counterfactual exhibits two failure modes: (i) inflated coverage due to borderline cases that our selective pipeline would escalate, or (ii) reduced coverage when trait scores cluster near 0.5.

These results underscore the value of calibrated dual thresholds: they preserve precision on passes, reveal cases with weak evidence, and support abstention as a principled fallback. Full diagnostics and comparisons appear in Appendix A.

## 4.6 ORGANIC CRS-61 CASE STUDY

To probe ecological validity, we apply BRIDGE to 61 organic conversational recommender sessions (CRS-61) collected from a deployed system, reusing the same Judge prompt and calibrated thresholds as in the synthetic experiments. Under strict session-level semantics, G1 passes 48/61 sessions (all assistant turns have non-null retrieval scores above a fixed cutoff), G3 passes 55/61 sessions (at least one aim-relevant dialog act appears anywhere in the session), and G4 passes 46/61 sessions under an "all turns must pass" tone/verbosity rule. For G2, every session has at least one non-contradictory turn relative to retrieved KB; across 537 assistant turns, the contradiction screen flags 128 potential claim–evidence conflicts, which we treat primarily as a per-turn diagnostic layered on top of G1 rather than as a hard session-level gate.

For this high-stakes profile, we therefore use strict G4 as the intake gate for CRS-61, admitting 46 dialogs to the Judge+Mapper lane. On these dialogs we conduct a prompt-paraphrase robustness study, re-evaluating each trait under three paraphrased Judge prompts. Trait-wise score ranges are moderate for Cognitive Support and Self-Reflection (median range around 0.2, maximum 0.4) and larger for Transparency (median 0.3, mean approximately 0.34, tail to 0.8). Despite this variance, pass/abstain decisions on CRS-61 are largely stable, with only a small fraction of dialogs changing status under prompt paraphrases.

For concreteness, Appendix A, Figure A.2 provides an "anatomy of a decision" example, tracing a single CRS-61 dialog through G1–G4, Judge trait scores, and the dual-threshold mapper to an abstention decision.

## 5 DISCUSSION

**What we establish.** A *human-anchored*, abstaining evaluation policy can deliver high-precision passes while routing a calibrated minority of dialogs to review. Dual thresholds fixed from a small human slice ($\alpha = 1 - \kappa$, centered on prevalence) induce interpretable *safety margins* that align abstentions with traits where humans themselves agree less. At the study operating point, this yields an 87.7% reduction in human workload with explicit, auditable reasons to escalate, without post hoc tuning.

---

[7]Shares computed over abstaining dialogs within each condition. Multiple traits may trigger escalation in the same dialog, so shares do not sum to 100%.

**Positioning and guarantees.** Selective prediction and deferral typically hinge on model confidence or a learned rejector (Geifman & El-Yaniv, 2017; 2019), and conformal methods offer distribution-free guarantees under proper calibration (Romano et al., 2020; Angelopoulos et al., 2024). BRIDGE *recenters abstention on human agreement*, using the Judge purely as a scoring instrument and mapping scores through thresholds derived from human prevalence and $\kappa$. This choice explicitly addresses recent evidence that LLM evaluators can recognize and prefer their own generations and are sensitive to prompt artifacts in survey-like settings (Panickssery et al., 2024; Dominguez-Olmedo et al., 2024). We make *process guarantees* rather than distribution-free ones: disjoint calibration, frozen thresholds, and deterministic mapping. When formal pass-risk control is required, BRIDGE can be wrapped with conformal risk control to target a user-specified pass risk $\alpha$ while leaving the escalation workflow unchanged. Closely related work formalizes selective judging with provable human-agreement guarantees and cascaded escalation, reinforcing our design choice (Jung et al., 2025). Modeling human preference diversity further motivates anchoring thresholds to agreement rather than raw judge confidence (Bakker et al., 2022).

**Learning to defer (optional).** Where expert traces exist, a learned router can layer on top of BRIDGE to optimize allocation between humans and models (Madras et al., 2018; Mozannar & Sontag, 2020; Mao et al., 2023). Emerging work on training LLMs as judges underscores both the opportunity and the need for auditable safeguards around judge reliability (Ye et al., 2025). Until such traces and validated judge training are available, the rule-based abstention is a robust default that already yields large reductions in human load and transparent rationales for review.

**Portability across tasks.** The interface (gates → Judge scores → human-anchored mapper) ports by swapping the *trait schema* and retracing the same calibration steps. This keeps operator workflow stable across domains (for example, code planning versus summarization) and preserves selective prediction behavior with trait-level explanations for every decision.

**Operations, ethics, and governance.** Because thresholds are fixed from a disjoint human slice and applied post hoc, operators can (i) audit consistency, since scores → labels are deterministic under frozen bounds, (ii) tune coverage *without retraining* by adjusting the operating point (cf. §4.5), and (iii) target staffing to the *specific traits* that drive abstention. In high stakes settings, this supports the policy "escalate rather than overclaim," translating uncertainty into a measurable review queue. The choice to abstain rather than to trust raw confidence is further motivated by calibration issues in modern networks (Guo et al., 2017; Minderer et al., 2021). Publishing frozen thresholds, trait-level counts, and figure or metric scripts (no raw dialogs) enables third parties to *recalibrate on private data* and verify that observed decisions match the declared rule. Because error profiles can shift across populations, the abstention queue provides a natural locus for ongoing fairness and safety monitoring.

**Limitations and scope.** Our primary testbed is controlled rather than organic, and our organic case study consists of only 61 CRS-61 sessions from a single deployed system, so generalization beyond these distributions is an empirical question. The human slice (40 dialogs, two raters) yields nontrivial confidence intervals for $\kappa$ and ICC; nevertheless, recomputing thresholds at confidence interval endpoints leaves headline pass or abstain rates and the risk-first trait ordering unchanged at reported precision. We do not enforce conformal prediction or conformal risk control guarantees in this study; those are compatible wrappers when needed. Our synthetic setup lacks user-level strata; subgroup analysis and fairness-aware abstention are left for future work.

**Outlook.** Two immediate extensions preserve the core principle *abstain explicitly, ground decisions in human signal, and keep the policy auditable*: (i) conformal risk control wrapped operation when explicit pass-risk targets are mandated (Angelopoulos et al., 2024), and (ii) learned deferral when expert traces become available (Madras et al., 2018; Mozannar & Sontag, 2020; Mao et al., 2023).

# 6 CONCLUSION

We cast aim adherence as a selective evaluation problem with deterministic gates, a probabilistic Judge, and a conformal inspired mapper that sets per trait dual thresholds from empirical prevalence

and interrater agreement ($\alpha=1-\kappa$). A deterministic aggregator then passes only if all relevant traits are `yes`; otherwise the system escalates. On the synthetic testbed, $N=5{,}960$ dialogs admitted by G1–G4 yield an operating point where the pipeline passes 87.7% without human review (12.3% abstain), with passes clearing nontrivial evidence and abstentions concentrated in semantically appropriate traits. An organic CRS-61 case study on 61 anonymized conversational recommender sessions reuses the same gates, Judge prompt, and frozen thresholds without re-tuning. Under strict session-level semantics, G1 passes 48/61 sessions, G3 passes 55/61, and a strict G4 intake rule (all turns satisfy tone/verbosity constraints) admits 46/61 sessions to the Judge+Mapper lane, while G2 serves primarily as a per-turn contradiction screen layered on retrieval. On these dialogs, trait scores show moderate variability under Judge prompt paraphrases—especially for Transparency— yet pass/abstain decisions are largely stable, with only a small fraction of dialogs changing status. Calibration shows the expected monotone relation between agreement and band width; thresholds are frozen with no post hoc tuning, and observed decisions match the declared rule within numerical precision. We report $\kappa$ and ICC confidence intervals on a small human slice and verify threshold sensitivity at the interval endpoints; headline pass or abstain rates and the risk first trait ordering are stable at reported precision, and lower $\kappa$ only increases conservatism. We make process guarantees, not coverage guarantees, and release frozen thresholds, summary tables, and figure and metric scripts with no raw dialogs so others can recalibrate on private data. Grounding abstention in human agreement and publishing an audit trail turns evaluation into a risk aware decision process that practitioners can operate and trust across both synthetic testbeds and organic logs.

## REPRODUCIBILITY STATEMENT

We release a minimal, reviewer-verifiable artifact that reproduces all reported gate-level and Judge-level aggregates *without external APIs or proprietary assets*. The package contains (i) frozen CSV tables for G1–G4 by condition (with salted–hashed dialog references for removals), (ii) per-dialog Judge outcomes and a summary matching Table 2, (iii) a one-click verifier that re-aggregates counts and confirms Judge intake $N=5{,}960$, and (iv) metadata for the calibrated thresholds (file checksum and rounding-consistent summaries). No prompts, KB passages, detector patterns, or raw trait probabilities are released. All computations are deterministic from the provided tables; figure scripts regenerate plots and tables solely from these artifacts. Data are licensed CC BY-NC 4.0; the small verifier scripts are MIT-licensed.

## ETHICS STATEMENT

Our main experiments evaluate conversational recommenders using synthetic dialogs with limited human review conducted under institutional policies. For the CRS-61 case study we additionally analyze 61 anonymized logs from a deployed system, stripped of personal identifiers and used under institutional policies. No raw CRS-61 dialogs are released, and materials avoid sensitive attributes. To mitigate risks, the evaluation employs calibrated abstention with human escalation for ambiguous cases; we report transparent counts and operating thresholds. We release code and aggregate artifacts (e.g., thresholds, figures, metric scripts) with clear licensing and usage guidance, while withholding full dialog corpora to reduce misuse, benchmark gaming, and unintended downstream training. This work complies with institutional governance requirements and the ICLR Code of Ethics.

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

## A ROBUSTNESS AND CONSISTENCY

**A.1 Safety margins.** Figure A.1 reports the distance from empirical human prevalence to the nearest threshold for each trait ("nearest-bound safety margin"). Smaller margins indicate higher sensitivity to base-rate drift and are handled conservatively by our abstention policy.

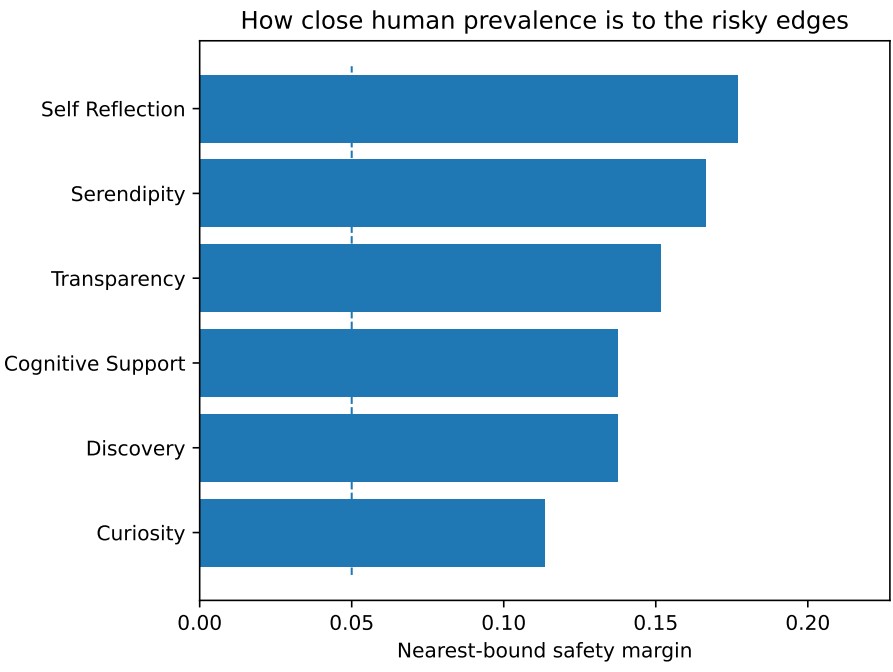

Figure A.1: **Nearest-bound safety margins by trait.** Distance from human prevalence to the closest threshold.

**A.2 Threshold consistency (no post-hoc tuning).** Stored thresholds match the declared construction $\mathrm{prev} \pm (1 - \kappa)/2$ (clipped to $[0, 1]$). Deviations are numerically zero on the calibration set (max $|\Delta| = 2.22e - 16$, median $|\Delta| = 6.94e - 17$).

**A.3 Numeric summary.** Agreement and band width are strongly related (Pearson $r \approx -0.99$; Spearman similar), consistent with the main-text scatter. Per-trait calibration (same order as figures): $\kappa$, prevalence, thresholds, band width, and safety margin.

Table A.1: Per-trait calibration metrics. For each trait, we compute empirical human prevalence (`prev`) and interrater agreement ($\kappa$) from a disjoint human slice ($n$=40 dialogs). Dual thresholds are computed as $\mathrm{Lower}, \mathrm{Upper} = \mathrm{clip}\left(\mathrm{prev} \pm \frac{1-\kappa}{2}, \, 0, \, 1\right)$, where $\mathrm{clip}(z, 0, 1) = \max\{0, \min\{1, z\}\}$. These fixed thresholds are applied post hoc during evaluation to map trait scores to {`yes`, `ambiguous`, `no`} decisions.

| Trait | $\kappa$ | Prev. | Lower | Upper | Width | Safety margin |
|---|---|---|---|---|---|---|
| Curiosity | 0.77 | 0.88 | 0.761 | 0.989 | 0.227 | 0.114 |
| Discovery | 0.69 | 0.86 | 0.707 | 1.000 | 0.293 | 0.137 |
| Cognitive Support | 0.72 | 0.84 | 0.700 | 0.975 | 0.275 | 0.137 |
| Transparency | 0.70 | 0.71 | 0.561 | 0.864 | 0.303 | 0.151 |
| Serendipity | 0.67 | 0.33 | 0.159 | 0.491 | 0.333 | 0.166 |
| Self Reflection | 0.65 | 0.44 | 0.261 | 0.615 | 0.354 | 0.177 |

**A.4 Sensitivity to tolerance $\alpha$.** Selective evaluation uses abstention to meet a target tolerance $\alpha$. Tightening $\alpha$ widens bands and increases escalation predictably; trait risk ranking is stable.

Table A.2: Sensitivity analysis of the abstention policy under stricter tolerance levels. We simulate alternative abstention behaviors by decreasing the threshold half-width parameter $\alpha = 1 - \kappa$ to $\alpha \in \{0.80, 0.60, 0.40\}$ (i.e., assuming lower human agreement). This tightens the `yes`/`no` bands and increases abstention. The table reports the predicted percentage of dialogs escalated under each $\alpha$ setting, using stored Judge scores. No dialogs were regenerated; all values reflect post hoc remapping.

| $\alpha$ | Trait | $\kappa$ | Prev. | Pred. escalation |
|---|---|---|---|---|
| 0.05 | Curiosity | 0.77 | 0.88 | 0.23 |
| 0.05 | Discovery | 0.69 | 0.86 | 0.29 |
| 0.05 | Cognitive Support | 0.72 | 0.84 | 0.28 |
| 0.05 | Transparency | 0.70 | 0.71 | 0.30 |
| 0.05 | Serendipity | 0.67 | 0.33 | 0.33 |
| 0.05 | Self Reflection | 0.65 | 0.44 | 0.35 |
| 0.10 | Curiosity | 0.77 | 0.88 | 0.11 |
| 0.10 | Discovery | 0.69 | 0.86 | 0.16 |
| 0.10 | Cognitive Support | 0.72 | 0.84 | 0.14 |
| 0.10 | Transparency | 0.70 | 0.71 | 0.15 |
| 0.10 | Serendipity | 0.67 | 0.33 | 0.17 |
| 0.10 | Self Reflection | 0.65 | 0.44 | 0.18 |
| 0.20 | Curiosity | 0.77 | 0.88 | 0.06 |
| 0.20 | Discovery | 0.69 | 0.86 | 0.08 |
| 0.20 | Cognitive Support | 0.72 | 0.84 | 0.07 |
| 0.20 | Transparency | 0.70 | 0.71 | 0.08 |
| 0.20 | Serendipity | 0.67 | 0.33 | 0.08 |
| 0.20 | Self Reflection | 0.65 | 0.44 | 0.09 |

**A.5 Threats to validity** Bands are derived once from a fixed human-annotated set; base-rate shifts can move operating points. We center bands on empirical prevalence, expose safety margins (A.1), for audit; no raw dialogs are released.

**A.6 Anatomy of a decision (CRS-61 example).** Figure A.2 illustrates how BRIDGE processes a single organic CRS-61 dialog through gates, Judge, and mapper. The example is schematic but follows an actual pattern from the CRS-61 study: a long assistant turn fails G4 under strict tone/verbosity semantics; for dialogs that pass gates, the Judge produces trait scores which are then mapped via frozen thresholds to YES/NO/AMBIGUOUS, and any non-YES trait triggers abstention.

# B EXPERIMENTAL SETUP

**B.1 Conditions and data.** Four cells: Travel–Educative, Travel–Explorative, Lifestyle–Educative, Lifestyle–Explorative; 1,500 dialogs per cell, each with six user and six assistant turns. Assistant responses reference KB chunks via simulated `retrieved_chunks`.

**B.2 Guardrails and judging.** G1–G4 filter invalid/unsafe/misaligned/overly-verbose items. The Judge assigns trait scores per dialog; a conformal mapper applies dual thresholds to output `yes`/`no`/`ambiguous`. Any `ambiguous`/`no` is routed to human review.

**B.3 Models and decoding.** Evaluated across a fixed temperature grid $\{0.0, 0.3, 0.5, 0.7, 1.0\}$ with shared max tokens and per-aim stop rules. Seeds and environment are disclosed in Appendix D ($\{13, 17, 23\}$, Python 3.11.x).

# C HUMAN AGREEMENT

Two trained annotators labeled the calibration set. Across traits, Cohen's $\kappa \approx 0.64$–$0.77$ and ICC(2,1) $\approx 0.56$–$0.87$. Thresholds are computed directly from prevalence and $\kappa$ with clipping and are frozen as released numeric artifacts.

**Gates (CRS-61 dialog)**

User/assistant turns (abbreviated):
U1: "I want book recommendations..."
A1: "Here are some options..."
U2: follow-up question
A2: long, digressive response (176 words)

G4 rule (strict):
```
turn_pass        =
(bad_tone        =
0) ∧ (long_turn = 0)
session_pass_G4  =
all_turns(turn_pass)
```

Here A2 exceeds the length threshold, so G4 fails and the dialog is routed to ABSTAIN_GATE (G4_TONE_BREACH).

**Judge scores and thresholds**

For dialogs that pass all gates, the Judge produces trait scores, e.g. for an educative aim:
Transparency $= 0.35$
Cognitive Support $= 0.62$
Self-Reflection $= 0.41$

Frozen thresholds (example):
Transparency:
Lower $= 0.56$,
Upper $= 0.78$
Cognitive Support:
Lower $= 0.40$,
Upper $= 0.70$
Self-Reflection:
Lower $= 0.45$,
Upper $= 0.69$

Thresholds are derived once from the human calibration slice and reused across synthetic and CRS-61 evaluations without re-tuning.

**Mapper and final decision**

Dual-threshold mapper (per trait $t$):
YES if $\hat{p}_t \geq \text{Upper}_t$;
NO if $\hat{p}_t \leq \text{Lower}_t$;
AMBIGUOUS otherwise.

Applying to the example scores:
Transparency $0.35 < 0.56$ $\Rightarrow$ NO
Cognitive Support $0.62$ in band $\Rightarrow$ AMBIGUOUS
Self-Reflection
$0.41 < 0.45 \Rightarrow$ NO

Because at least one trait is NO or AMBIGUOUS, the system abstains (ABSTAIN_MAPPER) for this dialog and surfaces it for human review; the aim-level fidelity is $F_a = \min_t \hat{p}_t$.

(a) Gating (G4 failure under strict tone/verbosity).

(b) Judge scores with human-anchored thresholds.

(c) Mapper labels and abstention.

Figure A.2: **Anatomy of a decision on CRS-61.** Schematic example showing (a) how a long assistant turn can fail G4 under strict tone/verbosity rules, (b) Judge trait scores with frozen thresholds derived from the human calibration slice, and (c) the dual-threshold mapper converting scores to YES/NO/AMBIGUOUS and abstaining when any trait is non-YES.

Table D.1: Summary of model generation and runtime configuration used in evaluation. Includes decoding parameters, hardware, Python environment, and experimental grid settings.

| | |
|---|---|
| Temperatures | {0.0, 0.3, 0.5, 0.7, 1.0} |
| Seeds | {13, 17, 23} |
| Python | 3.11.x |
| Environment | conda/venv (requirements pinned) |
| Hardware | Single GPU (A100) or CPU-only for judging |
| Decoding | Shared max tokens and per-aim stop rules (see code) |

## D    REPRODUCIBILITY AND RELEASE NOTES

- Code, figure scripts, and pinned environments are provided; runs record commit, seed, and hardware.
- Calibration artifacts are numeric only (thresholds, counts, tables); no raw dialogs are released.
- Licensing: paper under CC BY 4.0; datasets/non-text assets released separately with non-commercial terms.

