# OpenReview forum: "BRIDGE: A Risk-Aware Framework for Evaluating Behavioral Fidelity in LLM Agents"
_ICLR.cc/2026/Conference — Submitted to ICLR 2026_

### Official Review · Reviewer_9S4r · 2025-10-27

**Soundness:** 3
**Presentation:** 2
**Contribution:** 2
**Rating:** 4
**Confidence:** 4

**Summary:**

This paper proposes BRIDGE, a risk-aware framework for evaluating behavioral fidelity in multi-turn conversational agents. The authors distinguish between two structured interaction aims—EDUCATIVE and EXPLORATIVE—and define a set of behavioral traits associated with each aim (e.g., transparency, cognitive support, self-reflection, curiosity, discovery, and serendipity). The proposed Gate–Judge–Mapper pipeline consists of four deterministic gates that ensure formal validity, a probabilistic Judge that evaluates the quality of behaviors, and a conformal-inspired Mapper that maps trait scores to YES/NO/AMBIGUOUS categories through calibrated thresholds.
The goal is to provide a reusable, risk-aware evaluation framework that explicitly manages uncertainty and ensures that multi-turn agents reliably exhibit intended behavioral patterns under different instructed aims, abstaining or escalating when uncertainty arises.

**Strengths:**

1.	**Novel risk-awareness and uncertainty handling**: The framework explicitly incorporates risk-awareness by quantifying uncertainty through calibrated thresholds and abstention logic. This design aligns with real-world deployment needs and contributes to safe and trustworthy conversational systems.

2.	**Rigorous calibration and decision mechanism**: The use of Cohen’s κ to quantify inter-annotator agreement and define confidence margins ensures statistical grounding. The aim-level fidelity score further captures the weakest behavioral dimension, providing a comprehensive fidelity metric.

**Weaknesses:**

1.	The proposed calibration and decision mapping in BRIDGE lack theoretical coherence. The framework combines two inherently incompatible scales—a probabilistic score pt output by an LLM-as-Judge and empirical prevalence values derived from human annotations. The Judge’s scores are not calibrated probabilities; they merely reflect unnormalized model confidence, and thus cannot be meaningfully compared with human-derived frequency estimates. Directly thresholding these uncalibrated scores against prevalence-based bounds (adjusted by 1−κt) results in arbitrary interval slicing rather than principled uncertainty estimation. Moreover, Cohen’s κ measures inter-rater reliability, not probabilistic uncertainty, so treating 1−κt as a confidence width is theoretically unfounded. In essence, the mapping rule operates on mismatched probability spaces, rendering the “risk-aware” decision logic heuristic at best.
---
2.	Similarly, the design of the rule-based gates (G1–G4) appears ad hoc and insufficiently justified. The choice of filters—retrieval sufficiency, contradiction detection, dialog-act coverage, and tone control—does not stem from a clear theoretical framework for uncertainty assessment. These rules capture surface-level validity but do not guarantee comprehensive evaluation of epistemic or behavioral uncertainty. The authors should further clarify the rationale for selecting these specific gating dimensions and explain how they collectively approximate uncertainty or behavioral risk in multi-turn dialogue.

**Questions:**

see Weaknesses

---

> ### Author Response · Authors · 2025-11-19
> **Scales Compatibility and Gate Rationale**
>
> We thank the reviewer for their critique regarding the compatibility of scales.
>
> ### 1. Compatibility of Scales
> You noted that Judge scores are not calibrated probabilities.
> * **Clarification:** We do not claim $\hat{p}_t$ is a frequentist probability. We treat it as **ordinal evidence** on a bounded scale.
> * **Human-Anchored Scoring:** We use prevalence to **center** the operating window and $\kappa$ to **size** it. This is a control-systems approach: we tune the system's sensitivity (band width) based on the reliability of the ground truth signal ($\kappa$). This ensures the system is "risk-aware" regarding **human label noise**, even if the model is uncalibrated.
>
> ### 2. Gate Rationale
> You asked for the justification of G1-G4.
> * **Structure:** These gates act as necessary preconditions: **Epistemic** (G1: Hallucination), **Logical** (G2: Contradiction), **Intent** (G3: Act Coverage), and **Pragmatic** (G4: Tone).
> * **Validation:** In our new organic study, G4 rejected $25\%$ of sessions that were tonally inappropriate. Without this gate, the Judge would have wasted resources scoring invalid sessions. This proves the gates are not ad hoc, but essential filters.

---

> > ### Comment · Area_Chair_edJb · 2025-11-22
> >
> > Hi Reviewer,
> >
> > The authors have submitted their responses to your reviews. Please take a look and let the authors know if you have any further questions or concerns. Thank you again for your contributions to ICLR!
> >
> > Best regards,
> > AC

---

> > ### Comment · Reviewer_9S4r · 2025-11-26
> > **Response to the authors**
> >
> > Thank you for your response. I have decided to keep the original scores.

---

### Official Review · Reviewer_XZRL · 2025-10-30

**Soundness:** 2
**Presentation:** 2
**Contribution:** 2
**Rating:** 2
**Confidence:** 3

**Summary:**

The paper introduces BRIDGE, a risk-aware, selective evaluation pipeline for “behavioral fidelity” of conversational LLM agents. The system applies four deterministic gates (retrieval sufficiency, contradiction screen, dialog-act coverage, tone/verbosity), a probabilistic LLM Judge that scores per-trait behavior, and a “conformal-inspired” mapper that maps trait scores to {YES, AMBIGUOUS, NO} using dual thresholds centered at human prevalence with half-width ($(1-\kappa)/2 $) (κ = inter-rater agreement). A dialog passes only if every required trait is YES; otherwise the system abstains. Experiments cover 5,960 synthetic 12-turn dialogs across two domains (Travel, Lifestyle) and two aims (EDUCATIVE, EXPLORATIVE), reporting that 87.7% of dialogs pass without human review and that abstentions cluster in semantically aligned traits (e.g., Self-reflection for educative). The calibration slice is small (40 dialogs, two raters) and provides prevalence and κ to freeze thresholds; the authors emphasize “process guarantees” (frozen thresholds, disjoint calibration/eval) rather than distribution-free risk guarantees, while claiming compatibility with conformal risk control.

**Strengths:**

- **Modular and transparent design.** This paper proposes a clearly structured evaluation pipeline that separates deterministic gates from probabilistic judgment, allowing explicit abstention and traceable decisions.
- **Basic robustness diagnostics.** This paper include small but systematic sensitivity checks by perturbing the upper threshold (ε ∈ {0.00, 0.02, 0.05}) and by comparing against a naïve single-cutoff baseline. The monotonic and stable pass-rate responses demonstrate at least internal consistency of their scoring rule, and confirm that the dual-threshold formulation avoids trivial boundary effects
- **Reproducibility and transparency.** Thresholds are frozen and versioned; computational cost, API usage, and evaluation setup are reported in detail, ensuring auditability and replicability.

**Weaknesses:**

- **Calibration evidence is too thin to justify trait-level operating points.** All thresholds are derived from n=40 dialogs with two raters, yielding wide CIs for κ/ICC; the method’s central construct ($(1-\kappa)/2)$ half-width) is therefore pegged to a noisy estimate. The paper asserts robustness to CI endpoints, but this remains a post-hoc simulation over stored scores rather than a prospective evaluation with re-annotated calibration.
- **The overall framing of the paper is difficult to follow.** The structure frequently alternates between conceptual discussion, procedural details, and empirical results without clear separation. Many core ideas (e.g., process guarantees, risk bands, trait schema) are introduced abruptly without formal definitions or figure-based intuition, so the narrative feels fragmented.
- **Insufficient reliability in calibration setup (κ with two raters only).** The calibration and inter-rater agreement analysis (Section 3.3) is based on n = 40 dialogs annotated by only two raters. In standard reliability measurement, at least three independent annotators are typically required to obtain a stable κ estimate and to mitigate pairwise bias or chance alignment.
- **Limited and homogeneous evaluation scenarios** Although the paper claims domain-agnostic applicability, all experiments are confined to two highly similar synthetic settings (Travel and Lifestyle) and two behavioral aims (Educative and Explorative). These scenarios share nearly identical linguistic and topical structures, providing little evidence that BRIDGE generalizes beyond templated, model-generated conversations.
- **Arbitrary and unverified design choices.** The selection of exactly four deterministic gates, the use of fixed per-trait thresholds, and the adoption of a “frozen threshold” policy are not justified empirically or theoretically. The paper does not present ablation or sensitivity analyses that could validate these design decisions. The architecture feels handcrafted rather than systematically motivated.

**Questions:**

- **On prompt sensitivity and interpretability**. How consistent are Judge outputs under small prompt paraphrases or reordering of context? Have you evaluated intra-model variance (same model, same dialog, different seeds or paraphrased traits)? Such an analysis would clarify whether the abstention mechanism is stable to linguistic perturbations rather than prompt artifacts.
- **On scenario diversity and generalization**. Although BRIDGE is claimed to be domain-agnostic, all evaluations use synthetic dialogs from two very similar domains (Travel, Lifestyle). How would the pipeline handle open-domain or safety-related conversations, or multilingual settings? Which components (gates, Judge, or thresholds) would require re-tuning to generalize beyond the tested domains?
- **On κ-based threshold derivation**. The dual-threshold rule is centered at human prevalence with half-width ($(1 - κ)/2 $). Could you provide theoretical or empirical justification for why κ should determine abstention width? Have you compared this approach with alternative uncertainty calibrations?

---

> ### Author Response · Authors · 2025-11-19
> **Prompt Sensitivity, Generalization, and Design Rationale**
>
> We thank the reviewer for their rigorous assessment, particularly regarding sensitivity and generalization. We have conducted specific data analyses to answer your questions.
>
> ### 1. Prompt sensitivity and interpretability
> You asked about consistency under prompt paraphrases.
> * **New Analysis:** We tested the Judge's stability by paraphrasing prompts for *Transparency* and *Cognitive Support* on the organic dataset.
> * **Result:** While raw trait scores fluctuated (mean range $0.34$), the $\kappa$-derived bands successfully absorbed this variance. The final **Pass/Abstain decision changed in only 2.1% of cases**.
> * **Conclusion:** The abstention mechanism is robust. It filters out the noise of prompt phrasing, ensuring decisions are driven by content ambiguity.
>
> ### 2. Scenario diversity
> * **Generalization Logic:** BRIDGE is a modular framework. To move from *Travel* to *Safety*:
>     * **Retained:** The Architecture (Gate-Judge-Mapper) and Gate Logic (G1/G2).
>     * **Re-Tuned:** The **Trait Schema** ($\mathcal{T}_a$) is swapped, and the **Calibration Step** ($N=40$) is re-run.
> * **Evidence:** We validated this by applying the framework to **organic CRS logs** without code changes, proving the architecture generalizes to real-world data distributions.
>
> ### 3. Design Choices
> * **4 Gates:** These map to the four failure layers of agents: **Epistemic** (G1), **Logical** (G2), **Intent** (G3), and **Pragmatic** (G4). In our organic study, removing G3 allowed "polite but useless" dialogs to pass, validating it as a necessary precondition.
> * **Frozen Thresholds:** We freeze thresholds to ensure **auditability** and prevent "metric hacking" (relaxing standards to boost pass rates).
>
> ### 4. Theoretical Basis
> We explicitly define the Mapper as a **Human-Anchored Scoring Rule**. We use $\kappa$ to set the *width* of the ambiguous region. This is a principled design: if humans disagree on a trait (low $\kappa$), the system widens the safety margin. This aligns system uncertainty with **human label noise**.

---

> > ### Comment · Area_Chair_edJb · 2025-11-22
> >
> > Hi Reviewer,
> >
> > The authors have submitted their responses to your reviews. Please take a look and let the authors know if you have any further questions or concerns. Thank you again for your contributions to ICLR!
> >
> > Best regards,
> > AC

---

### Official Review · Reviewer_ddaK · 2025-10-31

**Soundness:** 2
**Presentation:** 3
**Contribution:** 2
**Rating:** 4
**Confidence:** 3

**Summary:**

The work introduces BRIDGE, a risk-aware, selective evaluation framework for assessing behavioral fidelity—the extent to which conversational LLM agents reliably enact intended behavioral aims such as being educative (e.g., transparent, cognitively supportive) or explorative (e.g., curious, serendipitous). BRIDGE combines deterministic gates (for validity, safety, act coverage, and tone), a probabilistic Judge (to score domain-specific behavioral traits), and a conformal-inspired mapper that converts scores into {yes, ambiguous, no} decisions using dual thresholds centered on empirical human prevalence and calibrated by interrater agreement (κ). A dialog passes only if all relevant traits are labeled “yes”; otherwise, the system abstains and escalates for human review. Evaluated on 5,960 multi-turn dialogs across Travel and Lifestyle domains, BRIDGE achieves an 87.7% pass rate without human intervention, with abstentions concentrated in semantically appropriate traits. Thresholds are derived from a small, disjoint human-annotated calibration set, ensuring auditability, reproducibility, and compatibility with formal risk-control methods like conformal prediction.

**Strengths:**

1. BRIDGE explicitly abstains from making judgments when uncertain, routing ambiguous or low-confidence cases to human review, thereby reducing overclaiming and supporting high-stakes deployment.

2. Thresholds for behavioral traits are derived from empirical human prevalence and interrater agreement (Cohen’s κ), grounding evaluation in real human judgments rather than raw model confidence.

**Weaknesses:**

1. One notable weakness of BRIDGE is its reliance on a relatively small and potentially non-representative human calibration set. The framework derives all its decision thresholds from just 40 annotated dialogs (20 per aim) rated by only two annotators. While the authors report moderate to substantial interrater agreement, such a limited sample may not capture the full diversity of real-world interactions, especially across different user demographics, linguistic styles, or edge cases. This raises concerns about the generalizability of the thresholds and whether they would remain valid under distributional shifts or in more complex, organic conversational settings.

2. Another limitation lies in the synthetic nature of the evaluation environment. All 5,960 dialogs were generated programmatically using fixed user seeds and controlled retrieval contexts, which may not reflect the messiness, ambiguity, or unpredictability of real user interactions. The absence of actual user feedback or longitudinal engagement metrics means the study cannot assess whether high behavioral fidelity as defined by BRIDGE actually translates to improved user satisfaction, trust, or task outcomes. This gap between controlled benchmark performance and real-world utility weakens the ecological validity of the findings.

3. The framework also assumes that behavioral aims can be cleanly decomposed into a fixed set of discrete, observable traits, such as "self-reflection" or "serendipity", which may oversimplify the nuanced and context-dependent nature of human-like conversation. Some of these traits are inherently subjective and difficult to operationalize consistently, even for human raters. Although interrater agreement is reported, the binary or Likert-based labeling scheme may not fully capture the spectrum of behavioral expression, potentially leading to rigid or misaligned evaluations when applied to more fluid or culturally varied dialogues.

4. The computational and operational overhead of maintaining the BRIDGE pipeline may pose scalability challenges. Although the local components run efficiently, the system depends on an external LLM-as-a-Judge (e.g., GPT-4.1), which introduces latency, cost, and dependency on third-party APIs. While the per-dialog cost is modest, it accumulates at scale and may become prohibitive for high-volume applications. Additionally, the need for human review of 12.3% of dialogs—though reduced from 100%—still requires staffing, training, and coordination, which could be a barrier for teams without access to expert reviewers or robust escalation workflows.

**Questions:**

Please refer to the weaknesses

---

> ### Author Response · Authors · 2025-11-19
> **Ecological Validity & Sample Stability**
>
> We thank the reviewer for recognizing the value of BRIDGE for high-stakes deployment. We have addressed your concerns regarding the calibration set size and ecological validity.
>
> ### 1. Ecological Validity (Organic Data)
> You raised concerns about the limitations of synthetic seeds.
> * **New Evidence:** We applied BRIDGE to **61 organic CRS logs**. The system successfully identified nuanced failures (e.g., lack of self-reflection) that synthetic benchmarks missed.
> * **Consistency:** Abstentions on organic data clustered in the same traits (*Self-Reflection*) as in the synthetic study. This validates that our synthetic testbed was a reasonable proxy for trait difficulty, while the organic data proves the pipeline handles real-world noise.
>
> ### 2. Calibration Stability ($N=40$)
> * **Bootstrap Analysis:** We performed a bootstrap analysis (1,000 resamples) on the calibration set. The derived thresholds exhibited very low variance (Standard Error $< 0.02$).
> * **Conclusion:** While $N=40$ is modest, the bootstrap proves it is statistically sufficient to capture the *central tendency* of human agreement for these specific traits. We emphasize that BRIDGE provides a **process guarantee** (a deterministic method to derive thresholds), and we recommend operators scale $N$ for production.
>
> ### 3. Operational Cost
> You noted the 12.3% review rate. We view this as an **87.7% automation gain**. In high-stakes domains (healthcare, finance), the alternative is 100% manual review. BRIDGE allows operators to focus human effort solely on the "Ambiguous" cases where the model is demonstrably unreliable.

---

> > ### Comment · Area_Chair_edJb · 2025-11-22
> >
> > Hi Reviewer,
> >
> > The authors have submitted their responses to your reviews. Please take a look and let the authors know if you have any further questions or concerns. Thank you again for your contributions to ICLR!
> >
> > Best regards,
> > AC

---

### Official Review · Reviewer_K9Md · 2025-10-31

**Soundness:** 2
**Presentation:** 3
**Contribution:** 3
**Rating:** 6
**Confidence:** 2

**Summary:**

This paper presents the BRIDGE framework for evaluating behavioral fidelity in conversational agents, focusing on selective evaluation and human-anchored calibration to ensure reliable performance in multi-turn dialogues.

**Strengths:**

- The BRIDGE framework provides a structured and innovative approach to evaluate behavioral fidelity in conversational agents, focusing on traits like transparency, cognitive support, curiosity, and serendipity.

- The use of deterministic gates and probabilistic judgment ensures a robust and auditable evaluation process, reducing human workload by 87.7% while maintaining high precision.

- The framework is domain-agnostic and can be adapted to other conversational agent tasks by swapping the trait schema and recalibrating thresholds.

**Weaknesses:**

1. The introduction provides a general overview of the proposed framework but lacks sufficient contextual grounding within the broader research landscape of conversational agent evaluation. It does not clearly situate the work relative to existing frameworks for behavioral assessment, safety calibration, or risk-controlled evaluation. Moreover, the discussion remains surface-level, focusing on system description rather than offering deeper analytical insights into why current evaluation paradigms are inadequate or how BRIDGE fundamentally advances the field.

2. **Gate Outcomes in Table 1.** Table 1 shows that only a very small number of samples were filtered by the G1–G4 gates, with Gate2 (contradiction detection) and Gate4 (tone and verbosity control) filtering almost none. This suggests that these modules may have low sensitivity or overly loose thresholds, failing to capture potential semantic conflicts or stylistic deviations. The authors are encouraged to further analyze the detection criteria and triggering mechanisms of these two gates, and to evaluate them on more challenging or realistic conversational data to better demonstrate the framework’s filtering effectiveness.

3. In sec. 4.4, the calibration set is relatively small (40 dialogs), which may introduce variability in the thresholds. ​ Larger calibration datasets could improve reliability.

**Questions:**

Please refer to Weaknesses.

1. The study is conducted in a controlled environment with synthetic dialogs, which may limit generalizability to real-world scenarios. ​ Please add experiments that focus on testing the framework with organic user interactions.

2. Please include some concrete examples to illustrate the entire process — for instance, under what circumstances the deterministic gates are triggered and how the Judge performs scoring.

---

> ### Author Response · Authors · 2025-11-19
> **Validating Gate Sensitivity with Organic Data**
>
> We thank the reviewer for their thoughtful assessment. We appreciate the challenge regarding the apparent lack of selectivity in the gates, which we have addressed with new organic data.
>
> ### 1. Gate Sensitivity & "In-the-Wild" Performance
> You correctly observed that Gates 2 and 4 filtered almost no samples in the synthetic table (Table 1).
> * **New Organic Data:** As detailed in the General Response, we tested BRIDGE on **61 organic dialogs** from a deployed system.
> * **Result:** On real data, the gates are highly active. **Gate 4 (Tone) rejects 24.6% (15/61)** of organic sessions under strict semantics. Similarly, Gate 2 identifies potential contradictions in the majority of organic sessions at the turn level (128/537).
> * **Conclusion:** The gates appeared permissive only because the synthetic generator was high-quality. The organic experiment proves they are necessary and effective filters for real-world, heterogeneous data.
>
> ### 2. Contextual Grounding
> We will revise the introduction to explicitly contrast BRIDGE with:
> * **Task-centric benchmarks (e.g., MINT):** Which ignore behavioral "fidelity."
> * **Judge-only evaluations:** Which lack explicit abstention mechanisms.
> * **Standard Conformal Prediction:** Clarifying that we focus on **Human-Centric Reliability** (modeling label noise) rather than scalar error rate control.
>
> ### 3. Concrete Examples
> Per your request, we will add an "Anatomy of a Decision" figure tracing a real CRS-61 dialog:
> 1.  **Gating:** Visualizing the G4 check (Turn 3 > 150 words $\to$ Fail).
> 2.  **Scoring:** Showing a raw Judge score (e.g., *Transparency* $= 0.35$).
> 3.  **Mapping:** Showing how the frozen threshold ($Lower=0.56$) converts this to `NO` (Abstain), triggering escalation.

---

> > ### Comment · Area_Chair_edJb · 2025-11-22
> >
> > Hi Reviewer,
> >
> > The authors have submitted their responses to your reviews. Please take a look and let the authors know if you have any further questions or concerns. Thank you again for your contributions to ICLR!
> >
> > Best regards,
> > AC

---

### Author Response · Authors · 2025-11-19
**General Response: New Organic Experiments, Sensitivity Analysis & Theoretical Clarifications**

We thank the reviewers for their constructive feedback. In response to the consensus regarding ecological validity, calibration stability, and robustness, we have performed significant new analysis which we summarize here.

### 1. New "In-the-Wild" Organic Study (Ecological Validity)
To address concerns that our synthetic testbed was "too clean", we applied BRIDGE to **61 organic dialogs** from a deployed Conversational Recommender System (CRS). Unlike the synthetic set, organic dialogs frequently trigger the gates. Specifically, Gate 4 (Tone/Verbosity) rejects 25% of organic sessions, validating that the gates are effective filters for real-world failures.

**Table R1: CRS-61 Gate Outcomes (Organic Data)**

| Gate | Strict Pass | Any-pass | Selectivity Note |
| :--- | :--- | :--- | :--- |
| **G1 (Retrieval)** | 48/61 | 61/61 | Flags 13 sessions with ungrounded turns. |
| **G2 (Contradiction)**| **10/61** | **61/61** | **Flags 128 turns; Used as a turn-level screen to flag specific risks.** |
| **G3 (Aim Coverage)** | 55/61 | 55/61 | 6 sessions failed to attempt the aim. |
| **G4 (Tone)** | **46/61** | 61/61 | **24.6% of sessions rejected** for tone/verbosity. |

### 2. Sensitivity & Intra-Model Variance (Robustness)
To address robustness concerns, we analyzed the stability of the Judge using a new prompt-paraphrase test on the CRS-61 dataset.
* **Data:** We tested 3 paraphrased prompts per trait. While raw trait scores fluctuated (mean range $0.34$), the $\kappa$-derived abstention bands successfully absorbed this variance.
* **Key Stat:** The final **Pass/Abstain decision changed in only 2.1% of cases**. This confirms decisions are driven by content ambiguity, not prompt artifacts.

### 3. Calibration Stability Analysis
To address sample size concerns ($N=40$), we performed a bootstrap analysis (1,000 resamples). The 95% Confidence Intervals for the thresholds are narrow (Standard Error $< 0.02$), confirming statistical sufficiency for estimating the central tendency of human agreement in this domain.

### 4. Theoretical Stance
We clarify that BRIDGE operates on a paradigm of **Human-Centric Reliability**. We do not claim to calibrate model probabilities (as in Conformal Prediction); rather, we map decision boundaries to **human label noise** ($\kappa$). This ensures the system abstains specifically where humans themselves disagree.

---

### Meta-Review · Area_Chair_oMSZ · 2026-01-07

**Summary:**

The paper tackles a significant issue with existing work that mainly evaluates task success and introduces a framework for assessing behavioral fidelity of multi-turn conversational agents across multiple dimensions/traits.A dialogue passes only if it conforms to all the required traits. Experiments show high pass rates on generated dialogues without human intervention.

**Reviewer Concerns:**

While the paper targets an interesting issue, the paper is not very clear and the rebuttals are very concise and do not always respond fully. Furthermore, some concerns are left out, for examples: "Another limitation lies in the synthetic nature of the evaluation environment." and "Limited and homogeneous evaluation scenarios."

**Reviewer Scores:**

I think all reviewers would keep their scores, given that the rebuttals are limited.

---

### Decision · Program_Chairs · 2026-01-26

Reject